

# A daily, 1-km resolution dataset of downscaled Greenland ice sheet surface mass balance (1958-2015)

Brice Noël[1], Willem Jan van de Berg[1], Horst Machguth[2,3,4], Stef Lhermitte[5], Ian Howat[6], Xavier Fettweis[7], and Michiel R. van den Broeke[1]

[1]Institute for Marine and Atmospheric research Utrecht, Utrecht University, Utrecht, Netherlands.
[2]Department of Geography, University of Zurich, Zurich Switzerland.
[3]Department of Geosciences, University of Fribourg, Fribourg, Switzerland.
[4]Geological Survey of Denmark and Greenland GEUS, Copenhagen Denmark.
[5]KU Leuven, Department of Earth & Environmental Sciences, Leuven, Belgium.
[6]Byrd Polar Research Center and School of Earth Sciences, Ohio State University, Columbus, USA.
[7]Department of Geography, University of Liège, Liège, Belgium.

*Correspondence to:* Brice Noël (B.P.Y.Noel@uu.nl)

**Abstract.**

This study presents a dataset of daily, 1-km resolution Greenland ice sheet (GrIS) surface mass balance (SMB) covering the period 1958-2015. Using elevation dependence, the high-resolution product is statistically downscaled from the native daily output of the polar regional climate model RACMO2.3 at 11-km. The dataset includes all individual SMB components projected to a down-sampled version of the Greenland Ice Mapping Project (GIMP) digital elevation model and ice mask. The 1-km mask better resolves narrow ablation zones, valley glaciers, fjords and disconnected ice caps. Relative to the 11-km product, the more detailed representation of confined glaciated areas leads to increased precipitation over the southeastern GrIS. In addition, the downscaled product shows a significant increase in runoff owing to better resolved low-lying marginal glaciated regions. The combined corrections for elevation and bare ice albedo markedly improve model agreement with a newly compiled dataset of ablation measurements.





## 1 Introduction

During the last two decades, the Greenland ice sheet (GrIS) experienced significant mass
loss as a result of increased meltwater runoff and sustained high solid ice discharge from
marine-terminating outlet glaciers (Van den Broeke et al., 2009; Rignot et al., 2008, 2011;
Sasgen et al., 2012; Shepherd et al., 2012; Enderlin et al., 2014). To fill spatial and temporal
gaps in the scarce in-situ observations, regional climate models (RCMs) are often used to
produce maps of the GrIS surface mass balance (SMB; Van Angelen et al. (2013); Burgess
et al. (2010); Ettema et al. (2010a,b); Fettweis (2007); Fettweis et al. (2005, 2011); Noël
et al. (2015); Lucas-Picher et al. (2012)). RCMs explicitly calculate the individual SMB
components (Lenaerts et al., 2012), i.e. precipitation, runoff and sublimation, over the en-
tire ice sheet (Fig. 1) at high spatial and temporal resolution and over extended periods.
However, the current spatial resolution of RCMs, typically 5 to 20 km, remains too coarse
to accurately resolve glaciated areas in topographically complex regions such as small iso-
lated ice caps and marginal outlet glaciers flowing into narrow fjords. In these regions, the
relatively coarse elevation and land ice masks used in RCMs might result in runoff underes-
timation (Franco et al., 2012; Noël et al., 2015), hampering realistic regional SMB estimates.
Performing higher-resolution simulations to address these issues would require a substantial
computational effort and is thus restricted to case studies of small regions and relatively
short time periods.

As an alternative, statistical downscaling can be applied to RCM output. Previously,
this method has been applied to the GrIS using global reanalysis and climate data (Hanna
et al., 2005, 2008, 2011). Machguth et al. (2013) downscaled near-surface temperature and
precipitation from 3 different RCMs (11-25 km spatial resolution) to force a glacier mass
balance model on a 250 m grid derived from the Greenland Ice Mapping Project (GIMP)
digital elevation model (DEM) (Howat et al., 2014), accurately resolving local glaciers and
ice caps of Greenland. Vertical gradients of climate parameters were iteratively calibrated to
enable the mass balance model to generate a realistic melt distribution for the period 1980-
2010, but the very high resolution restricted the analysis to a few regions. Franco et al.
(2012) statistically downscaled GrIS SMB by interpolating each component of the Modèle
Atmosphérique Régional (MAR) from the original 25 km grid to a 15 km resolution. This
method used local daily vertical gradients, except for precipitation, to correct for elevation
differences between MAR and a down-sampled version of the 5 km DEM from Bamber et al.
(2001). The elevation correction significantly reduced SMB biases. However, a resolution
of 15 km remains insufficient to resolve the rugged topography at the ice sheet margins; to
address this issue, near-km resolution is necessary.

Here, we present a new dataset of daily, 1-km resolution GrIS SMB components (pre-
cipitation, melt, runoff, refreezing, sublimation and snowdrift erosion) covering the period





1958-2015. The SMB product is statistically downscaled from data of the polar regional climate model RACMO2.3 at 11-km (Fig. 1), using an elevation dependent technique based on the elevation and ice mask from the GIMP DEM (Howat et al., 2014), down-sampled to 1-km. The following section briefly describes RACMO2.3, the GIMP DEM, observational
datasets and MODIS bare ice albedo product used to evaluate and correct the downscaled dataset. The downscaling algorithm is explained in Section 3. Downscaled SMB is evaluated using ablation and accumulation measurements in Section 4. Section 5 discusses the downscaling results for four different regions and for the entire ice sheet, followed by conclusions in Section 6.

## 2   Model and data

### 2.1   The regional climate model RACMO2

A detailed description of the Regional Atmospheric Climate Model (RACMO2) is presented in Van Meijgaard et al. (2008). RACMO2 incorporates the atmospheric dynamics and physics modules from the High Resolution Limited Area Model (HIRLAM) and the Euro-
pean Centre for Medium-range Weather Forecasts Integrated Forecast System (ECMWF-IFS, Undèn et al. (2002)). The polar version of RACMO2 is developed by the Institute for Marine and Atmospheric Research (IMAU), Utrecht University, and is especially adapted for use over ice sheets and other glaciated regions. Polar RACMO2 is interactively coupled to a multi-layer snow module, accounting for firn densification, meltwater percolation, re-
freezing and runoff (Ettema et al., 2010a); an albedo scheme with prognostic snow grain size (Kuipers Munneke et al., 2011) and a drifting snow module, simulating snow erosion and the drifting snow contribution to sublimation (Lenaerts et al., 2012). Recently, RACMO2.1 has been updated to RACMO2.3 as discussed in Van Wessem et al. (2014) and Noël et al. (2015). Model evaluation against SMB measurements, collected in the accumulation and
ablation zones of the GrIS, showed generally improved agreement (Noël et al., 2015). The native 11-km climate run is forced at the lateral boundaries by ERA-40 (1958-1978, Uppala et al. (2005)) and ERA-Interim (1979-2015, Stark et al. (2007); Dee et al. (2011)) reanalyses and uses the 5 km DEM and ice mask from Bamber et al. (2001).

### 2.2   GIMP DEM

To downscale RACMO2.3 output, we use the ice mask and topography from the GIMP DEM, described in Howat et al. (2014), and currently considered to be one of the most complete ice masks for Greenland (Rastner et al., 2012). A 1-km ice mask and DEM are obtained by averaging the original 90 m GIMP grid-cells in each 1-km pixel covering Greenland. A 1-km resolution is deemed an acceptable trade-off between improved resolution, i.e. a 121





fold improvement compared to the 11-km grid, and manageable data handling given the daily time resolution, time span (1958-2015) and the number of SMB components. As an example, Figure 2a shows the topography and ice mask from RACMO2.3 at 11-km in central east Greenland (blue box 1 in Fig. 1) and Figure 2b the GIMP DEM at 1-km. The latter much better resolves small scale landforms such as narrow fjords and calving glacier tongues.

Integrated over the contiguous GrIS, the ice-covered area of $1.69 \; 10^6$ km$^2$ for the 1-km grid represents a 0.5% decrease relative to the 11-km mask. For our SMB calculations, we only consider grounded ice, i.e. we discarded floating ice pixels using a 1-km version of the 90 m grounded ice mask used in Enderlin and Howat (2013).

### 2.3   Ablation and accumulation measurements

To evaluate the daily downscaled SMB product, we use 1155 SMB measurements collected in the GrIS ablation (1073) and accumulation (182) zones. The ablation dataset (Machguth et al., in press/in revision) was compiled as part of the Programme for Monitoring of the Greenland Ice Sheet (PROMICE) (Van As et al., 2011) and includes stake and AWS measurements retrieved from 230 sites (yellow dots in Fig. 1). Accumulation observations were

derived from 182 sites including snow pits and firn cores (Bales et al., 2001, 2009) as well as airborne radar measurements (Overly et al., 2015) (white dots in Fig. 1). We exclusively selected data having a temporal overlap with RACMO2.3 simulations (1958-2015). We rejected observations from sites with a > 100 m height bias relative to the representative elevation of the 1-km GIMP topography.

To compare modelled and downscaled SMB with observations, different selection approaches were applied in the ablation and accumulation zones, as described in Noël et al. (2015). In the accumulation zone, we select the closest grid-cell on the 11-km and 1-km grids to represent modelled and downscaled SMB, respectively. In the ablation zone, an altitude correction is applied by selecting the grid-cell with the smallest elevation bias among the

closest pixel and its eight adjacent neighbours.

### 2.4   MODIS bare ice albedo

A 1-km version of the 500 m MODerate-resolution Imaging Spectroradiometer (MODIS) 16-day Albedo product (MCD43A3) is used to retrieve estimates of bare ice albedo in the GrIS ablation zone. Bare ice albedo is estimated as the average of the 5% lowest surface

albedo measurements for the period 2000-2015. A similar ice albedo product is used in RACMO2.3 based on MODIS observations between 2001 and 2010 (Noël et al., 2015). In RACMO2.3, bare ice albedo ranges from 0.3, i.e. dark bare ice exposed in the low ablation zone, to 0.55 under persistent snow cover in the GrIS accumulation zone. Bare ice albedo of glaciated pixels with no valid MODIS estimate are set to 0.47.





## 3   Methods

The daily, 1-km SMB product consists of statistically downscaled output from a previously
conducted RACMO2.3 simulation at 11-km, covering the period 1958-2015. RACMO2.3
settings and lateral forcing are described in Noël et al. (2015). The downscaling algorithm
corrects the interpolated SMB components using their local regression to elevation. Figure 3
shows the spatial correlation of individual SMB components with elevation on the 11-km
RACMO2.3 grid. The spatial correlation is calculated for each grid-box using 8 adjacent
ice-covered pixels.

The elevation correction is exclusively applied to the SMB components which show a signif-
icant and spatially homogeneous correlation with elevation, i.e. melt, runoff and sublimation
(Fig. 3). These SMB components decrease with decreasing air temperature, represented by
a negative correlation with elevation (Fig. 3b, d and e). Although precipitation negatively
correlates with elevation over most of the ice sheet, the correlation remains small and highly
heterogeneous at the margins (Fig. 3a). Snowdrift erosion exhibits a noisy correlation pat-
tern. Therefore, daily precipitation and snowdrift erosion are bi-linearly interpolated to the
1-km ice mask without elevation corrections. Refreezing exhibits a marked bimodal cor-
relation pattern (not shown), gradually increasing with height in the ablation zone, where
pore space is more abundant, and decreasing towards the ice sheet interior due to limited
meltwater supply. For this reason, and in order to have a consistent liquid water balance,
daily refreezing is calculated as a residual:

$$RF = RA + ME - RU \quad (1)$$

where RF is the residual refreezing, RA is rainfall, ME is surface melt, and RU is melt-
water runoff.

Daily SMB values are obtained by summing the individually downscaled components:

$$SMB = P_{tot} - RU - SU - ER \quad (2)$$

where $P_{tot}$ is total precipitation (liquid and solid), RU is meltwater runoff, SU is total
sublimation (from surface and drifting snow) and ER is drifting snow erosion.

### 3.1   Elevation dependent downscaling

The downscaling algorithm interpolates daily SMB components to the 1-km topography and
ice mask in three successive steps (Fig. 4a).



First, the local dependence on elevation is calculated on the original RACMO2.3 11-km grid. Regression parameters are computed on a daily basis and are, therefore, only valid for that specific day. A local regression slope, $b_{11km}$ (mmWE per m, Fig. 4a), is calculated for each ice-covered RACMO2.3 grid-point using at least 6 adjacent ice-covered pixels including the current one. This number is chosen after testing the downscaling sensitivity to the

number of regression cells used, as discussed in Section 3.2. An approximation of the SMB components at mean sea level, $a_{11km}$ (mmWE, Fig. 4a), is then obtained using $b_{11km}$ and the current pixel. Local regression parameters for melt and runoff are only computed for pixels experiencing ablation. Moreover, erratic positive regression slopes, i.e. increasing melt rates with altitude, are discarded until the following stage.

Next, valid estimates of $b_{11km}$ and $a_{11km}$ are extrapolated iteratively on the 11-km grid to fully cover the 1-km ice mask. To that end, regression parameters are extrapolated outwards of the 11-km ice mask by averaging $b_{11km}$ from at least 3 ice-covered pixels from the eight cells surrounding the current one.

    Finally, the extrapolated fields of $b_{11km}$ and $a_{11km}$ are bi-linearly interpolated to the

1-km ice mask, providing estimates of $b_{1km}$ and $a_{1km}$. The downscaled SMB components ($X_{v0.2}$), i.e. runoff, melt and sublimation, are then computed as a linear function of the high-resolution topography as:

$$X_{v0.2} = a_{1km} + b_{1km} \times elevation_{1km} \quad (3)$$


    The downscaled dataset that is based on the above elevation dependent technique is hereafter referred to as version v0.2.

### 3.2   Sensitivity experiment

    Figure 5 shows the difference between 11-km and downscaled, GrIS integrated daily runoff

in summer 2011. Each line represents a different number of grid-cells, ranging from 3 to 9, used to estimate the local regression of runoff with elevation (Fig. 4a). The results are moderately sensitive to the number of regression points used except for the 9 cells setting (current pixel and its 8 neighbours). The latter systematically underestimates runoff at the beginning and the end of the melt season as it discards all low-lying glaciated pixels at the

edge of the GrIS, which experience early melt and largest values of runoff. The standard deviation between the different settings ($\sim 0.2$ Gt/day) is significantly smaller than the difference between 11-km and 1-km runoff ($\sim 0.6$ Gt/day). The more regression points are used, the smoother the runoff to elevation gradient field becomes, lowering the downscaled runoff and bringing it closer to the 11-km model output. Conversely, a small number of

regression points can lead to spuriously large local gradients. To prevent the downscaling



algorithm from substantially converging to, or diverging away from, RACMO2.3 output, we adopted a setting of 6 regression points, which is closest to the average value of the different experiments ($\pm$ 0.1 Gt/day).

### 3.3   Melt and runoff adjustments

RACMO2.3 uses a prescribed bare ice albedo field, typically ranging from 0.30 in the low ablation zone to 0.55 under persistent snow cover. It is based on the 5% lowest MODIS values of surface albedo averaged for the period 2001-2010 (Noël et al., 2015). A comparison with a similar 1-km MODIS product averaged for 2000-2015, ranging from 0.15 to 0.55, shows a systematic overestimation of ice albedo at 11-km, especially for low-lying marginal

glacier tongues as shown in e.g. Fig. 12i and j. This causes melt energy to be underestimated during the melt season. To correct for this, downscaled melt and runoff are adjusted by estimating the missing amount of ice melt ($ME_{add}$) resulting from underestimated absorption of downward shortwave radiation ($SW_d$). In addition, as RACMO2.3 calculates radiative fluxes on a horizontal plane, the direct fraction of $SW_d$ is corrected for the slope

and orientation of each 1-km glaciated grid-cell, as described in Weiser et al. (2016). For simplicity, we assume $SW_d$ to be equally partitioned between diffuse and direct radiation, and that the sun is exactly in the South at noon. The following corrections are only applied when both surface runoff and melt are nonzero in the downscaled product (v0.2):

$$ME_{add} = \Delta\alpha \times 0.5 \left( \frac{SW_{d\ 1-km}}{L_f} + \xi \frac{SW_{d\ 1-km}}{L_f} \right) \quad (4)$$

where $ME_{add}$ (mmWE per day) is the additional amount of ice melt calculated at 1-km; $\Delta\alpha$ (-) is the difference between the averaged bare ice albedo retrieved from the set of regression cells used to downscale runoff at 11-km and the MODIS albedo product at 1-km;

$SW_{d\ 1-km}$ is the modelled daily cumulated downward shortwave radiation bi-linearly interpolated to 1-km; $L_f$ is the latent heat of fusion (3.337 $10^5$ J/kg) and $\xi$ (-) is the correction factor for a tilted plane, applied to the direct component of downward shortwave radiation:

$$\xi = \frac{cos(\zeta^*)}{cos(\zeta)}$$


$$\zeta^* = \sin(\zeta)\cos(a)\cos(\sigma)\cos(\Theta) + \sin(\zeta)\sin(\sigma)\sin(\Theta)$$

$$+ \cos(\zeta)\cos(\sigma)$$

$$\zeta = \text{acos}\Big(sin(\phi)sin(\delta) + cos(H)cos(\phi)cos(\delta)\Big) \quad (5)$$





where $\zeta^*$ is the solar angle of incidence for a tilted plane, $\zeta$ is the solar zenith angle, $a$ is the azimuth of the tilted plane, $\sigma$ is the local surface slope, $\Theta$ is the orientation, $\phi$ is the latitude, $\delta$ is the solar declination and H is the hour angle set to 0 at noon (Fig. 4b). All
angles are expressed in radians.

Additional runoff $RU_{add}$ is calculated by applying a daily specific fraction $\Gamma$ (-) to $ME_{add}$, estimating the melt contribution to surface runoff. $\Gamma$ is defined as the ratio between daily downscaled runoff and melt in v0.2 estimated using elevation dependence only:


$$RU_{add} = \Gamma \times ME_{add} \quad (6)$$

Assuming that the residual misfit between reconstructed and observed SMB ($\Delta$SMB, Fig. 6b) for the different ablation sites can be ascribed to underestimated runoff in the low ablation
zone of the GrIS, $RU_{add}$ is then scaled by a factor $f_{scale}$ (-), obtained by computing a least-square fit minimising the difference between $\Delta$SMB and $RU_{add}$ using all ablation measurements:

$$\Delta SMB = f_{scale} \times RU_{add}$$


$$f_{scale} = \frac{\sum \Delta SMB \times RU_{add}}{\sum (RU_{add})^2} \quad (7)$$

The least square fit yields a value of $f_{scale} = 1.176$ for the GrIS. The fact that $f_{scale} > 1$ suggests that additional processes might play a role in enhancing surface ablation, e.g.
underestimation of modelled sensible heat flux from warm air advection along the GrIS periphery (Noël et al., 2015; Fausto et al., 2016) and uncertainties in cloud representation (Van Tricht et al., 2016). The adjusted amount of runoff ($RU_{v1.0}$) is obtained by adding the missing runoff to the downscaled runoff ($RU_{v0.2}$).

$$RU_{v1.0} = RU_{v0.2} + f_{scale} \times RU_{add} \quad (8)$$

The corrected melt ($ME_{v1.0}$) is obtained in a similar way and refreezing ($RF_{v1.0}$) is estimated as a residual between adjusted melt, runoff and rainfall:

$$ME_{v1.0} = ME_{v0.2} + ME_{add} \quad (9)$$

$$RF_{v1.0} = RA + ME_{v1.0} - RU_{v1.0} \quad (10)$$





The downscaled SMB dataset resulting from the combined elevation correction and runoff
adjustment is referred to as version v1.0 in the following sections.

## 4   Evaluation of daily downscaled SMB

Figure 6 evaluates the original RACMO2.3 SMB at 11-km (a), the 1-km raw downscaled
SMB version v0.2 (b) and the 1-km corrected downscaled SMB version v1.0 (c) (mWE per
year) with 1073 observations from 230 ablation sites (yellow dots in Fig. 1). The observa-
tional period was matched with the modelled and downscaled SMB using the exact number
of days. Each blue star corresponds to the cumulative SMB for a duration ranging from 10
days to a full hydrological year. The downscaled SMB v0.2 agrees better with observations
compared to the RACMO2.3 output at 11-km (Figs. 6a and b): we find a significant decrease
of the RMSE (130 mmWE or -18%) and a smaller bias (50 mmWE or -24%). The devia-
tion from unity of the regression slope decreases from 0.29 to 0.23 (-21%), and the variance
explained increases from 47% to 61%. When applying the bare ice albedo and local orienta-
tion corrections, we find further significant improvements relative to version v0.2 (Fig. 6c),
with now 78% of the variance explained and a significant decrease in RMSE (180 mmWE
or -27%) and bias (140 mmWE or -88%). Red stars represent data from PROMICE station
QAS_L (61.03°N, 46.85°W, 310 m.a.s.l; yellow dot in Fig. 11) situated in an extremely
narrow ablation zone (∼ 10 km) at the southwestern tip of Greenland. Here, modelled abla-
tion gradients at 11-km are strongly underestimated in RACMO2.3 and are only marginally
better resolved at 1-km. At this site, the additional corrections are especially important to
obtain agreement with observations.

Figure 7 compares annual mean observed and downscaled SMB (v1.0) along 8 different
SMB transects. There is good agreement for most transects, except for Helheim glacier
(66.41N, -38.34W). Here, the original RACMO2.3 at 11-km fails to reproduce the SMB spa-
tial variability, likely because of its peculiar climate conditions. Helheim glacier experiences
pronounced accumulation in winter, caused by persistent advection of moist air from the
southeast, whereas strong ablation occurs in summer.

In the accumulation zone, a small improvement is also found compared to v0.2 (Fig. 8),
but accumulation remains underestimated. The SMB bias and RMSE are reduced by 0.7
(-2%) and 1.8 mmWE (-3%) whereas the regression slope and variance explained remain
unchanged. In the accumulation zone, SMB is mostly driven by precipitation which is
bi-linearly interpolated to 1-km without elevation correction. In addition, changes in sub-
limation are small due to the relatively homogeneous topography of the ice sheet interior,
limiting SMB changes through downscaling. To eliminate the systematic negative SMB
bias of RACMO2.3 in the GrIS accumulation zone (-37.5 mmWE/yr, Fig. 8), we adjusted





daily total precipitation v0.2 over areas showing positive annual cumulative SMB in v1.0

$(\mathrm{SMB}_{\mathbf{v1.0}} > 0$ mmWE/yr$)$:

$$\mathrm{PR}_{\mathbf{v1.0}} = \mathrm{PR}_{\mathbf{v0.2}} + \frac{\mathbf{PR}_{\mathbf{v0.2}}}{\mathbf{PR}_{\mathbf{v0.2}}^{\mathbf{a}}} \times \sigma_{SMB} \quad (11)$$

where $\mathrm{PR}_{\mathbf{v1.0}}$ is the daily adjusted total precipitation v1.0, $\mathrm{PR}_{\mathbf{v0.2}}$ is the daily bi-linearly

interpolated total precipitation v0.2, $\mathrm{PR}_{\mathbf{v0.2}}^{\mathbf{a}}$ is the annual cumulative bi-linearly interpolated total precipitation v0.2 and $\sigma_{SMB}$ is the accumulation zone SMB bias in the downscaled product v1.0.

## 5   High-resolution SMB patterns: case studies

Table 1 lists annual mean modelled and downscaled SMB components (Gt per year) inte-

grated over four different regions (blue boxes in Fig. 1) as well as over the entire GrIS. These regions were selected for their specific climates, rough topography and narrow glaciated features which were not well resolved at 11-km. Figures 9, 10, 11 and 12 show the ice sheet mask for the selected regions at 11-km (red cells) and 1-km (orange cells) as well as peripheral glaciers and ice caps at 1-km (blue cells), the elevation bias between the 11-km and 1-km

DEMs, and the bare ice albedo field as prescribed in RACMO2.3 at 11-km as well as the 1-km MODIS product; the latter figures moreover show the main SMB components at both resolutions for the two downscaled products (v0.2 and v1.0). In the following sections, we discuss the impact of downscaling on regional SMB. Here, SMB components are exclusively integrated over the contiguous GrIS; the SMB of detached ice caps will be discussed in a

forthcoming paper.

### 5.1   Central east Greenland

Central east Greenland (blue box 1 in Fig. 1) is characterised by a large body of intercon- nected valley glaciers, mostly terminating in narrow glacial fjords. Figure 9a, e, i and j underline the inability of the 11-km mask to properly represent many glaciated areas, lo-

cal topography or bare ice albedo. In the 1-km mask, the ice covered area increases by $\sim$ 2% while the elevation bias can locally exceed 500 m over glacial valleys and small scale promontories (Tab. 1 and Fig. 9e); the average elevation bias is 80 m. These differences affect SMB in two ways. First, precipitation increases by 2.6 Gt/yr or 12% in v0.2 (Tab. 1 and Fig. 9b and f), exclusively caused by the expansion of glaciated area (no elevation cor-

rection is applied). Another 1.6 Gt/yr of precipitation is added in v1.0 to compensate for the systematic negative SMB bias in the GrIS accumulation zone, as discussed in Section 4. For both downscaling versions, changes in runoff mirror the elevation change between the





two resolutions (Fig. 9e), highlighting the high sensitivity of runoff to elevation. In version
v0.2, integrated runoff increases by 7.7 Gt/yr (Fig. 9c and g). Furthermore, Fig. 9i and j
reveal a systematic overestimation of bare ice albedo at 11-km. Correcting for this further
increases runoff over the glaciers tongues (Fig. 9k), accounting for $\sim 13$ Gt/yr of additional
runoff with respect to v0.2 (Tab. 1). Negligible changes in sublimation and drifting snow
are found (Tab. 1). As a consequence, integrated SMB on the 1-km mask decreases by 5.3
Gt/yr in version v0.2 (Fig. 9d and h) and by 16.6 Gt/yr in version v1.0 (Fig. 9l). This
analysis for central east Greenland demonstrates the importance of accurately reproducing
small scale topography and ice albedo to realistically capture local SMB variations.

### 5.2    Central west Greenland

The 11-km resolution DEM provides a reasonable representation of the wide, gently sloping
western ablation zone of the GrIS, where most glaciers are land-terminating. The north-
ern part of the selected area includes several marine-terminating glaciers which are better
represented at 1-km (Fig. 10d and h).

Owing to negligible difference in glaciated area, precipitation remains almost unchanged
for the two resolutions and versions ($\sim 15$ Gt/yr). In both downscaled versions, enhanced
runoff is mostly obtained over narrow, low-lying glaciers tongues and detached ice caps
(Fig. 10c, g and k) where most of the elevation and ice albedo biases are found (Fig. 10e, i
and j). On the ice sheet, the elevation correction increased runoff by about 1 Gt/yr (Fig. 10h)
while an additional $\sim 2$ Gt/yr (Fig. 10l) can be ascribed to the ice albedo correction (Tab. 1).

### 5.3    South Greenland

Southeast Greenland (blue box 3 in Fig. 1) is a rugged region (Fig. 10e), characterized by
multiple topographically-forced precipitation maxima (Fig. 11b and f) and narrow marginal
ablation zones (Fig. 11c, g and k). Similar to central east Greenland, the larger the glaciated
area (+6.5%, Fig. 11a) at 1-km enhances integrated precipitation by $\sim 6$ Gt/yr (+7%) in
v0.2 and 8.4 Gt/yr (+9%) in v1.0. Increased runoff (2.2 Gt/yr in version v0.2) at the
southern margins can be ascribed to additional melt production over the better resolved
narrow ablation zones (Fig. 11d and h) combined with a moderate mean elevation difference
($\sim 17$ m) between both resolutions. In v0.2, the ice mask expansion explains most of the
integrated SMB changes, leading to an overall mass gain of 3.3 Gt/yr.

Fig. 6b reveals considerable ablation underestimation in southern Greenland, expressed
as a systematic SMB bias of 2 to 4 mWE relative to measurements collected at PROMICE
station QAS_L (red dots in Fig. 6a). The main reason for this underestimation is that SMB
at this location is characterized by a rare combination of high snowfall and strong summer
melt. The extreme elevational SMB gradient that results over the narrow ablation zone is





then poorly captured at 11-km, and hence also poorly represented at 1-km.

The remaining ablation underestimation in v0.2 can be partly ascribed to an overestimated
bare ice albedo (0.47) prescribed in RACMO2.3 (Noël et al., 2015); observed albedo at
QAS_L frequently falls to 0.2 during the melt season (Fausto et al., 2016). As a result,
the additional bare ice albedo correction significantly improves runoff at station QAS_L
(Fig. 6c). Integrated over region 3, runoff increases by another $\sim$ 13 Gt/yr relative to v0.2
(Fig. 11l). The increased marginal mass loss leads to the expansion of the southern ablation
zone towards higher elevations (Fig. 11k and l), in line with local observations (Fig. 6c).

### 5.4  North Greenland

In north Greenland (blue box 4 in Fig. 1), the climate is dry, and most glaciers are marine-
terminating. The ice sheet surface is relatively smooth and homogeneous. The wide ablation
zone is reasonably well captured at 11-km, leading to a modest deviation in elevation ($\sim$ 43
m) (Fig. 12e). However, the ice-covered area decreases by $\sim$ 11% between both resolutions
as the 11-km grid contained erroneous floating glacier tongues (Fig. 12a). The ice area
reduction at 1-km affects precipitation (-0.8 Gt/yr) (Fig. 12b and f) and runoff (-3.1 Gt/yr)
(Fig. 12c and g), resulting in a small SMB increase (2.3 Gt/yr) in version v0.2 (Fig. 12d
and h). Large bare ice albedo discrepancies can be found on five major glaciers (Fig. 12i
and j) where runoff increases substantially ($\sim$ 2 Gt/yr) in version v1.0, further decreasing
the integrated SMB by 1.0 Gt/yr compared to v0.2 (Fig. 12k and l).

### 5.5  Greenland ice sheet

Although similar in area, the 1-km ice sheet mask better resolves peripheral glaciers at
the GrIS margin than RACMO2.3 at 11-km. GrIS integrated precipitation increases by
16.6 Gt/yr (+2%) in v0.2, most of which can be ascribed to ice area expansion in the
east (2.6 Gt/yr) and south of Greenland (5.8 Gt/yr), where precipitation is large. An
additional 56.2 Gt/yr (+8%) is obtained in v1.0 when correcting for the accumulation zone
SMB bias. The smooth topography of the ice sheet interior results in a small elevation
difference of 4 m between both resolutions. Significant elevation biases are mostly restricted
to peripheral glaciers and narrow ablation zones at the GrIS margins. As a result, runoff
increases by 13.6 Gt/yr (+5%) in version v0.2. Accounting for the bare ice albedo bias
in RACMO2.3 further increases runoff by 69.3 Gt/yr in version v1.0, leading to a much
improved agreement with ablation measurements. Of our selected areas, central east and
south Greenland contribute 25% and 18% to the total runoff increase in the downscaled
product v1.0 owing to the many low-lying glaciers tongues that can only be resolved at
1-km. Due to their smoother topography, north and centre west Greenland contribute much
less to the runoff change ($\sim$ 3% and 1%, respectively). Integrated over the contiguous ice





sheet, SMB is not significantly affected by the elevation dependence for which enhanced precipitation (16.6 Gt/yr) yearly balances the moderate increase in runoff (13.6 Gt/yr). In

contrast, the bare ice albedo and precipitation corrections substantially increase marginal runoff (82.9 Gt/yr) and accumulation (72.8 Gt/yr), resulting in a decrease of SMB of -11.1 Gt/yr (-3%) relative to the 11-km product.

## 6  Limitations and uncertainty

The downscaled SMB v1.0 is likely to be locally underestimated for three reasons: a) the bare

ice albedo correction is evenly applied to both snow covered and bare ice regions experiencing surface melt and runoff, as no relevant proxy, reflecting day-to-day snow coverage, could be derived from RACMO2.3. However, this issue should have a limited effect on the magnitude of downscaled melt and runoff since the albedo correction is most efficient in summer, when the snow cover of low-lying glaciers has likely melted; b) the MODIS ice albedo product at

1-km becomes less accurate at high latitudes, likely suffering from bare soil contamination resulting from mixed reflectance signals recorded in both the tundra and ice covered regions; c) the average 1-km MODIS ice albedo product for 2000-2015 used in the melt correction remains constant in time and might underestimate the bare ice albedo prior to 2000. These limitations underline the high sensitivity of the downscaled product to the input fields used

to initialize the downscaling procedure, i.e. RCM version used, the resulting modelled SMB components, bare ice albedo records, ablation measurements, topography and ice mask. The downscaled SMB v1.0 presents an estimated uncertainty of $\sim$ 6 Gt/yr in the GrIS ablation zone, which was estimated by integrating the SMB bias in v1.0 (30 mmWE, Fig. 6c) over the ablation zone of the contiguous ice sheet ($\sim$ 202.000 km$^2$).

## 425  7  Conclusions

The relatively coarse spatial resolution currently used in RCMs remains insufficient to properly resolve small scale variations in elevation and ice cover at the ice sheet margins, significantly affecting the calculation of melt and runoff. In the present study, we statistically downscale individual SMB components from RACMO2.3 at 11-km to a 1-km ice mask

and topography derived from the GIMP DEM, using a daily specific elevation dependence. Moreover, runoff and melt are corrected for biases in bare ice albedo in RACMO2.3. Precipitation and snowdrift erosion are bi-linearly interpolated without applying an elevation correction. Total precipitation is also adjusted to compensate for the dry accumulation bias of RACMO2.3 in the ice sheet interior. Downscaled daily SMB is then retrieved for the pe-

riod 1958-2015 by summing daily downscaled precipitation, runoff, sublimation and drifting snow erosion. An evaluation of the downscaled SMB product against observations, collected





both in the ablation and accumulation zones of the GrIS, shows improved agreement. In
the ablation zone, the variance explained by the downscaled product v1.0 increased by 31%
relative to the original RACMO2.3 11-km output, mainly through better resolved narrow
outlet glaciers at the GrIS margins.

Integrated over the GrIS, precipitation increased by 16.6 Gt/yr due to the larger glaciated
area in south and east Greenland at 1-km; an additional correction of 13.6 Gt/yr must
account for the accumulation bias in the ice sheet interior in RACMO2.3. Likewise, a 26.4
Gt/yr increase in runoff is attributed to elevation corrections on the 1-km topography and
another 69.3 Gt/yr extra runoff can be ascribed to underestimated bare ice albedo over
narrow outlet glaciers at the GrIS margins. A small area in central east Greenland alone,
characterized by multiple narrow glacier tongues poorly resolved at 11-km, accounts for ∼
25% of the total additional runoff.

*Acknowledgements.* B. Noël, W. J. van de Berg, and M. R. van den Broeke acknowledge support
from the Polar Programme of the Netherlands Organization for Scientific Research (NWO/ALW)
and the Netherlands Earth System Science Centre (NESSC). I. Howat and the GIMP project
are supported by the U.S. National Aeronautics and Space Administration (NASA). H. Machguth
acknowledges support from the Programme for Monitoring of the Greenland Ice Sheet (PROMICE),
funded by the Danish Energy Agency's (DANCEA) program.





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





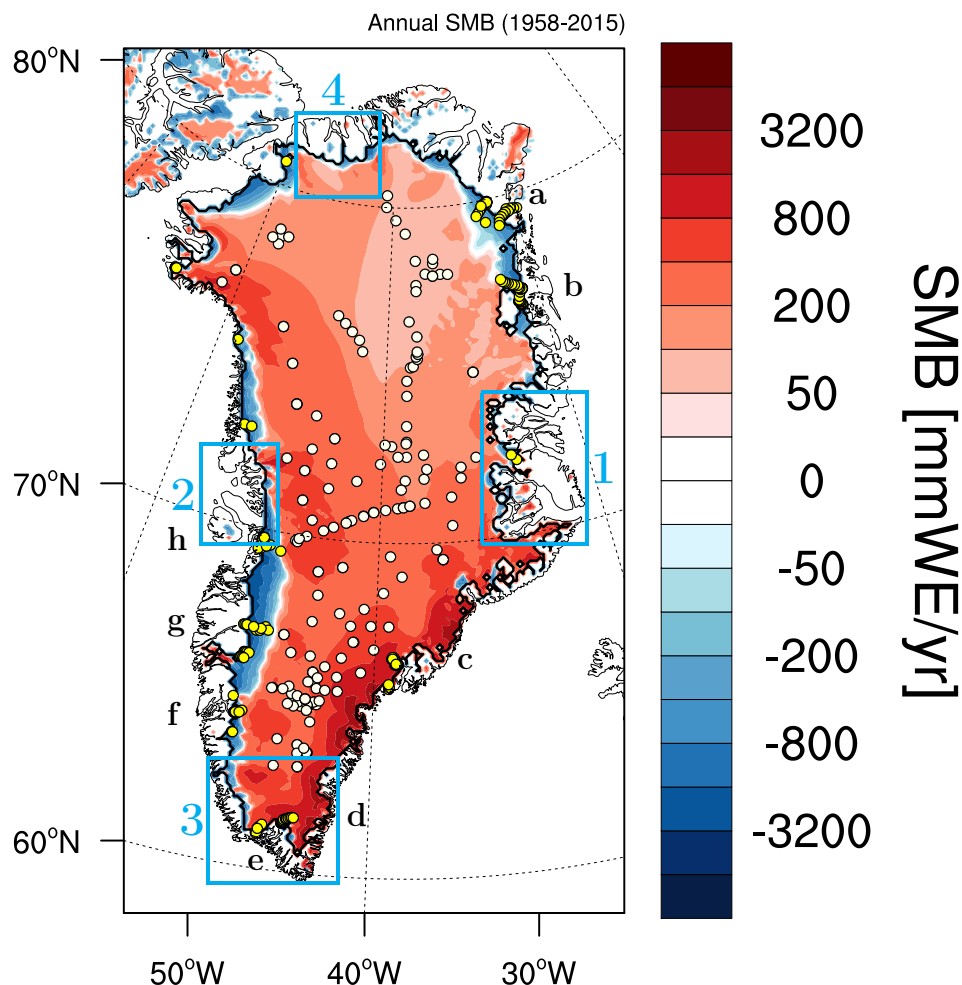

**Fig. 1.** Annual mean SMB modelled by RACMO2.3 at 11-km over the GrIS and surrounding ice caps for the period 1958-2015. This figure also depicts the location of 230 ablation measuring sites (yellow dots) and 182 accumulation sites (white dots) used for downscaled SMB evaluation as well as the four GrIS marginal regions (blue boxes), discussed in Section 5. Letters refer to the different transects shown in Fig.7.



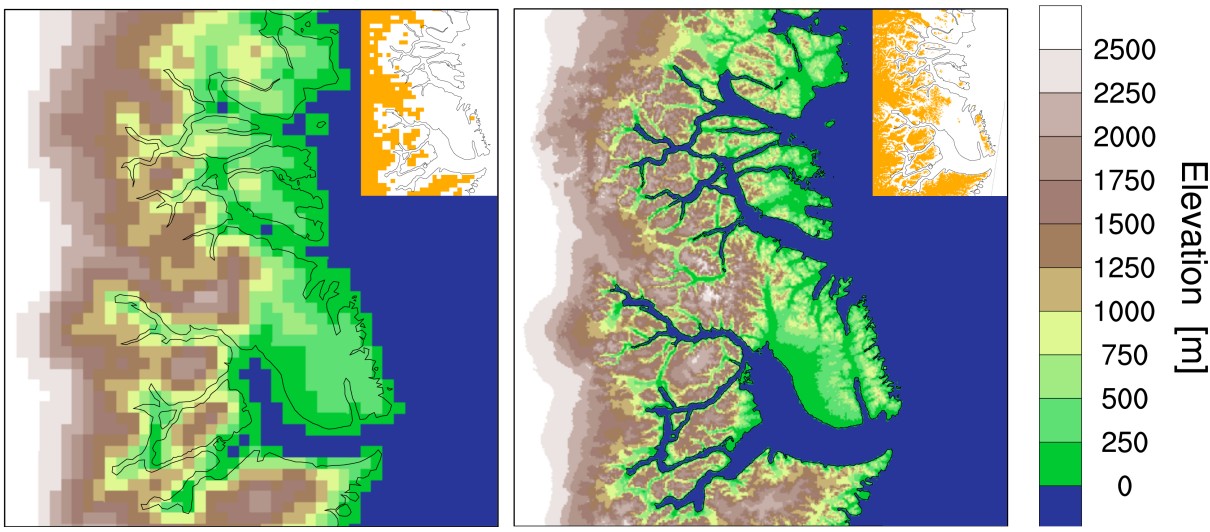

**Fig. 2.** Elevation and ice mask (yellow) as prescribed in RACMO2.3 at 11-km (left) and derived from the GIMP DEM down-sampled to 1-km (right) over central east Greenland (blue box 1 in Fig. 1).

| 1958-2015 | Regions | Centre east | | | Centre west | | | South | | | North | | | GrIS | | |
|---|---|---|---|---|---|---|---|---|---|---|---|---|---|---|---|---|
| **Resolution** | Unit | **11km** | **1km** | Δ | **11km** | **1km** | Δ | **11km** | **1km** | Δ | **11km** | **1km** | Δ | **11km** | **1km** | Δ |
| **SMB** $_{v0.2}$ | $Gt/yr$ | 5.0 | -0.3 | **-5.3** | -4.6 | -5.4 | **-0.8** | 44.3 | 47.6 | **3.3** | -2.6 | -0.3 | **2.3** | 349.3 | 351.3 | **2.0** |
| **Runoff** $_{v0.2}$ | $Gt/yr$ | 16.1 | 23.8 | **7.7** | 18.3 | 19.2 | **0.9** | 42.4 | 44.6 | **2.2** | 8.9 | 5.8 | **-3.1** | 284.1 | 297.7 | **13.6** |
| **Precip** $_{v0.2}$ | $Gt/yr$ | 22.6 | 25.2 | **2.6** | 15.0 | 15.2 | **0.2** | 91.4 | 97.2 | **5.8** | 6.9 | 6.1 | **-0.8** | 675.4 | 692.0 | **16.6** |
| **SMB** $_{v1.0}$ | $Gt/yr$ | 5.0 | -11.6 | **-16.6** | -4.6 | -6.7 | **-2.1** | 44.3 | 37.3 | **-7.0** | -2.6 | -1.3 | **1.3** | 349.3 | 338.2 | **-11.1** |
| **Runoff** $_{v1.0}$ | $Gt/yr$ | 16.1 | 36.7 | **20.6** | 18.3 | 21.1 | **2.8** | 42.4 | 57.5 | **15.1** | 8.9 | 7.7 | **-1.2** | 284.1 | 367.0 | **82.9** |
| **Precip** $_{v1.0}$ | $Gt/yr$ | 22.6 | 26.8 | **4.2** | 15.0 | 15.8 | **0.8** | 91.4 | 99.8 | **8.4** | 6.9 | 7.0 | **0.1** | 675.4 | 748.2 | **72.8** |
| **Sublimation** | $Gt/yr$ | 2.1 | 2.1 | **0.0** | 1.6 | 1.6 | **0.0** | 4.4 | 4.7 | **0.3** | 0.8 | 0.7 | **-0.1** | 41.3 | 41.9 | **0.6** |
| **Snow drift** | $Gt/yr$ | -0.5 | -0.4 | **0.1** | -0.3 | -0.2 | **0.1** | 0.2 | 0.3 | **0.1** | -0.1 | -0.1 | **0.0** | 0.7 | 1.1 | **0.4** |
| **Ice area** | $10^4$ km$^2$ | 5.9 | 6.0 | **0.1** | 2.7 | 2.7 | **-0.02** | 7.7 | 8.2 | **0.5** | 3.5 | 3.1 | **-0.4** | 170.3 | 169.4 | **-0.9** |

**Table 1.** Table listing (top) the annual mean integrated SMB components (Gt/year) covering the period 1958-2015 over four different regions, centre east (69.6°N – 74.3°N; 21°W – 31°W; blue box 1 in Fig. 1), centre west (69.3°N – 72.5°N; 49°W – 57°W; blue box 2), south (59.5°N – 63.3°N; 41°W – 51°W; blue box 3) and north (80.5°N – 83°N°; 42°W – 62°W; blue box 4), and for the entire GrIS at both resolutions as well as the difference between 1-km and 11-km; (bottom) same for the ice-covered area (km$^2$).




**Fig. 3.** Correlation to elevation of annual mean a) total precipitation (solid and liquid), b) runoff, c) SMB, d) sublimation, e) melt and f) drifting snow erosion modelled by RACMO2.3 and calculated on the 11-km grid for the period 1958-2015.



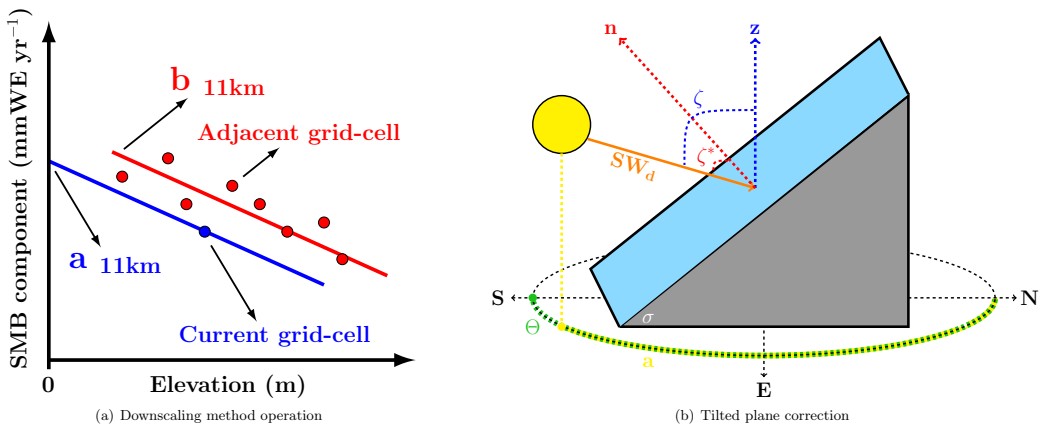

(a) Downscaling method operation      (b) Tilted plane correction

**Fig. 4.** (a) Elevation dependent downscaling procedure: local estimate of a daily SMB components regression to elevation on the RACMO2.3 grid at 11-km. (b) Scheme of a tilted plane as described in the GIMP DEM at 1-km.

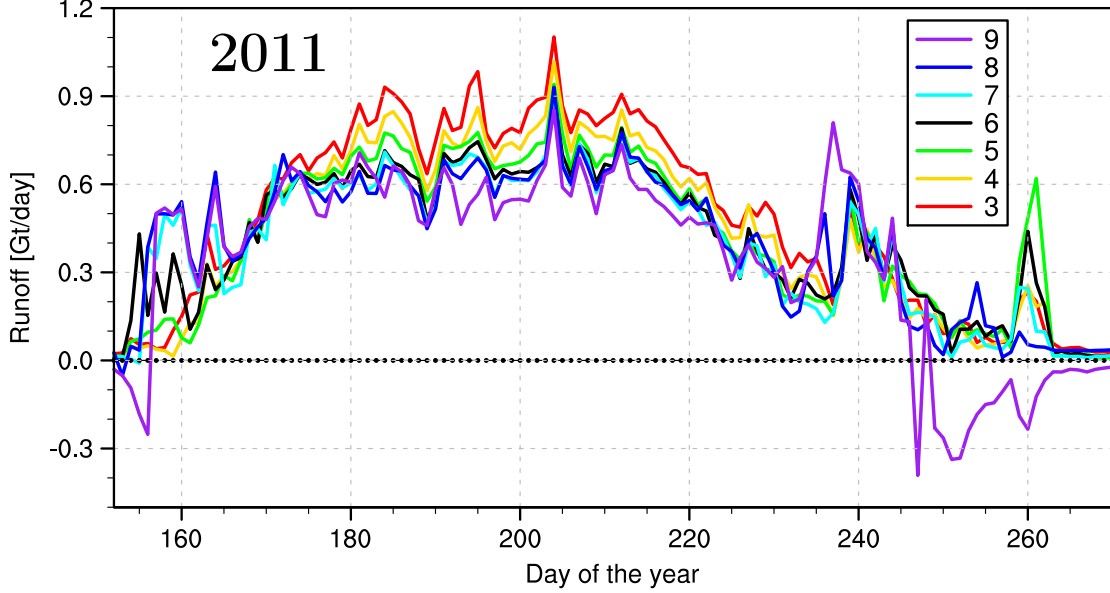

**Fig. 5.** Summer 2011 time series of daily, ice sheet integrated runoff difference (Gt/day) between the downscaled product at 1-km, using 3 to 9 regression points (legend), and the RACMO2.3 model at 11-km.





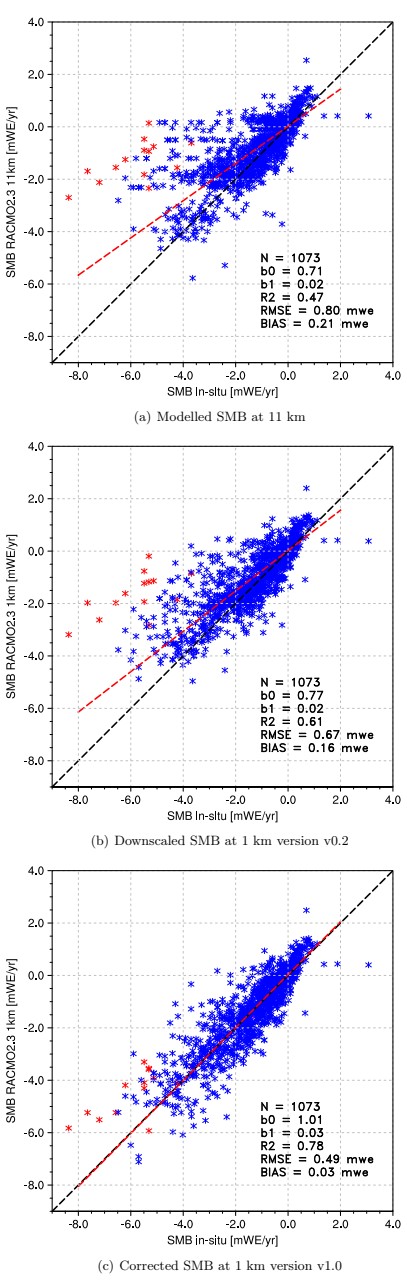

(a) Modelled SMB at 11 km

(b) Downscaled SMB at 1 km version v0.2

(c) Corrected SMB at 1 km version v1.0

**Fig. 6.** Comparison of SMB measurements collected at 230 sites with (a) modelled SMB from RACMO2.3 at 11-km; (b) downscaled SMB at 1-km (v0.2) and (c) corrected downscaled SMB at 1-km (v1.0). The red stars correspond to PROMICE station QAS_L located in southern Green-land (61.03°N, 46.85°W, 310 m.a.s.l). The red dashed line represents the regression including all measurements using a perpendicular fit.





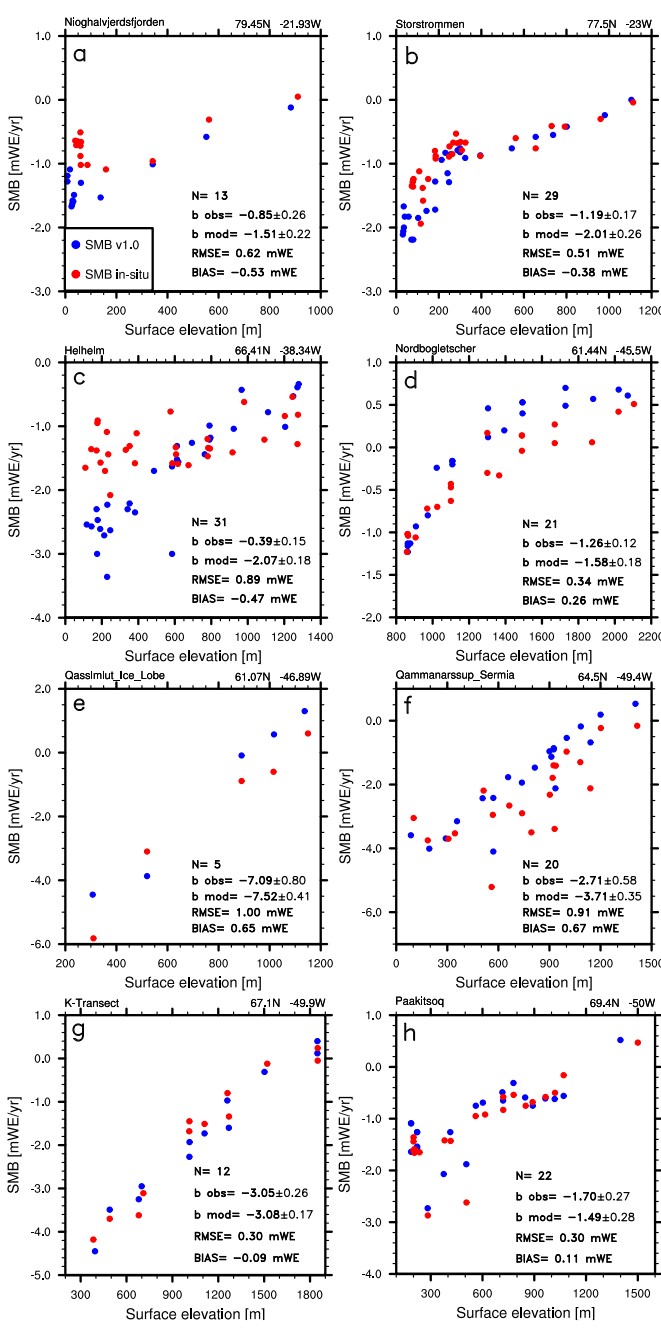

**Fig. 7.** Annual mean observed (red dots) and downscaled (blue dots, v1.0) SMB for 8 selected transects in the GrIS ablation zone (mWE/yr). Name and locations of these transects are listed at the top of each graph. Graphs also list the number of sites used for each transect, linear SMB-to-elevation regression retrieved from observations and downscaled (v1.0) data in mmWE/yr per m, RMSE and mean bias.





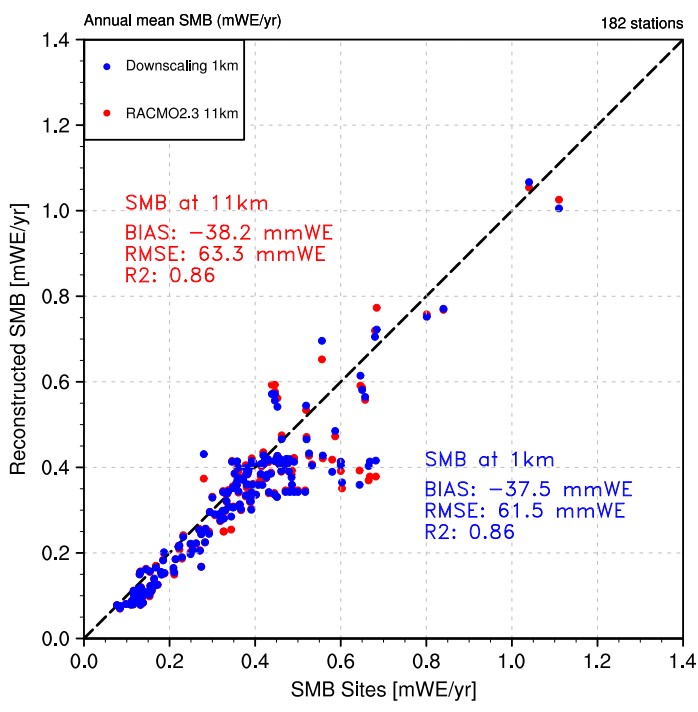

**Fig. 8.** Comparison of accumulation observations collected at 173 sites with modelled SMB from RACMO2.3 at 11-km (red) and downscaled SMB v1.0 at 1-km (blue) in mWE/yr. Note that bias correction has not yet been applied.



**Fig. 9.** a) Ice sheet mask in RACMO2.3 at 11-km (red) and in the down-sampled GIMP DEM at 1-km (orange) (blue box 1 in Fig. 1), and the local glaciers and ice caps mask at 1-km (blue); average (1958-2015) annual mean b) total precipitation, c) runoff and d) SMB (mmWE/yr) modelled by RACMO2.3 at 11km; e) elevation bias (m) between 1-km and 11-km resolutions. Figures f), g), h) represent annual mean downscaled total precipitation, runoff and SMB downscaled to 1-km using elevation dependence only (v0.2). Figures i) and j) show the bare ice albedo field as prescribed in RACMO2.3 at 11-km (2001-2010) and derived from MODIS measurements at 1-km (2000-2015). Figures k) and l) are similar to g) and h) but incorporate the bare ice albedo correction within the downscaling procedure (v1.0).





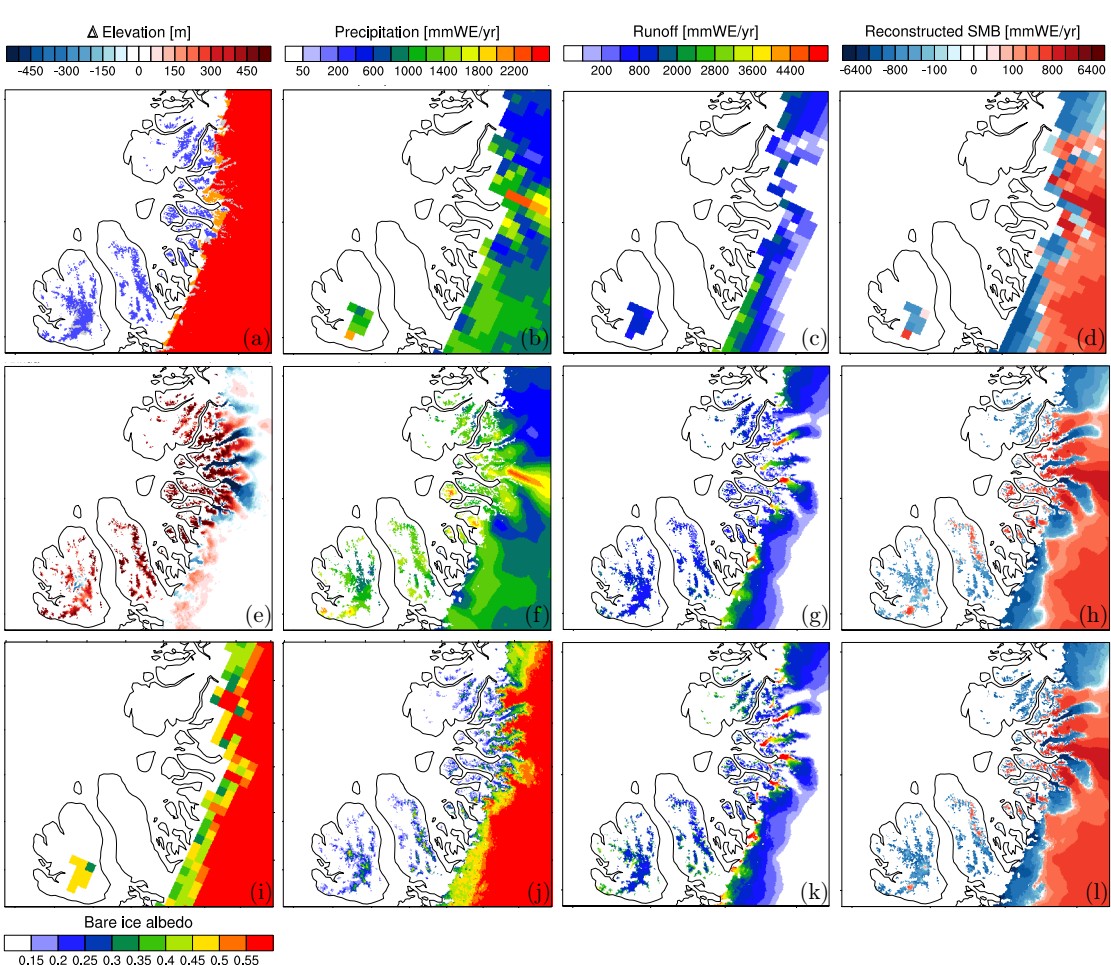

**Fig. 10.** Same as Fig. 9 but for central west Greenland (blue box 2 in Fig. 1).





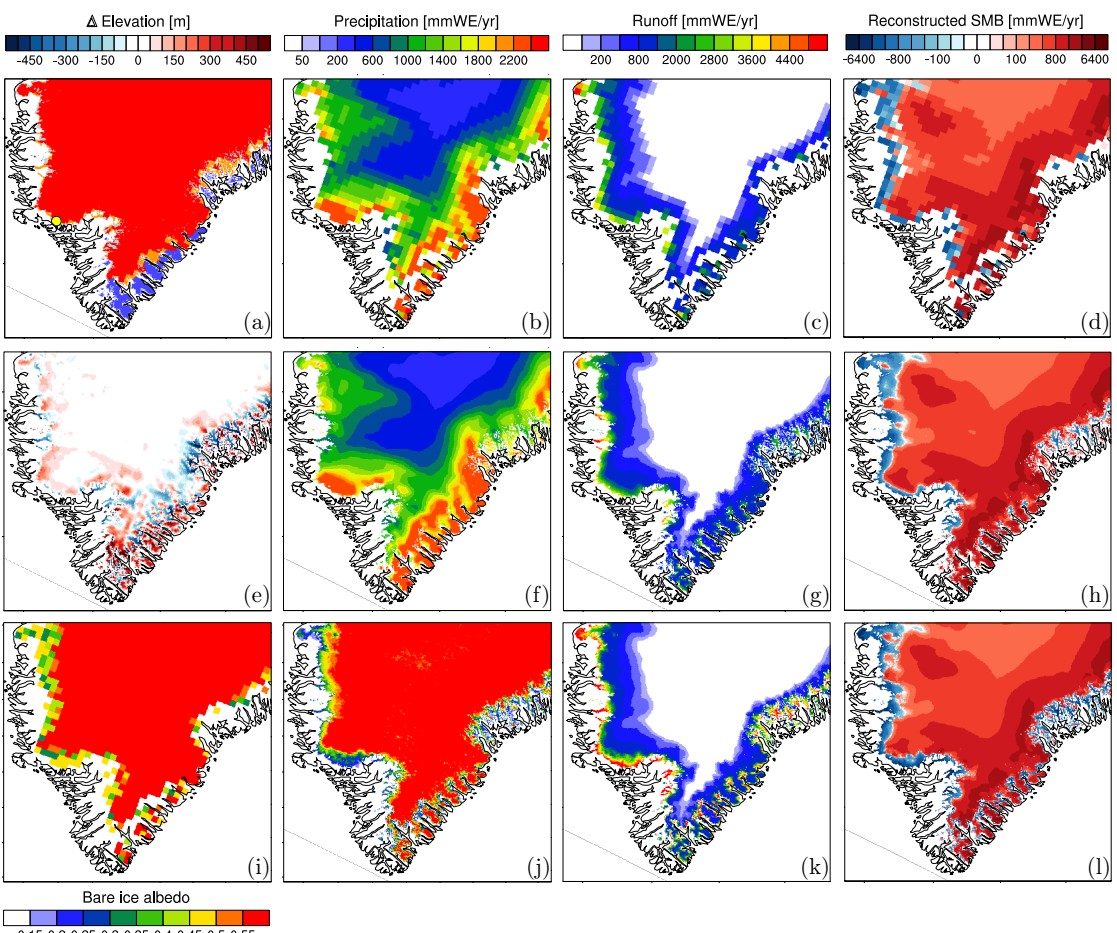

**Fig. 11.** Same as Fig. 9 but for south Greenland (blue box 3 in Fig. 1). The yellow dot in a) locates station QAS_L.





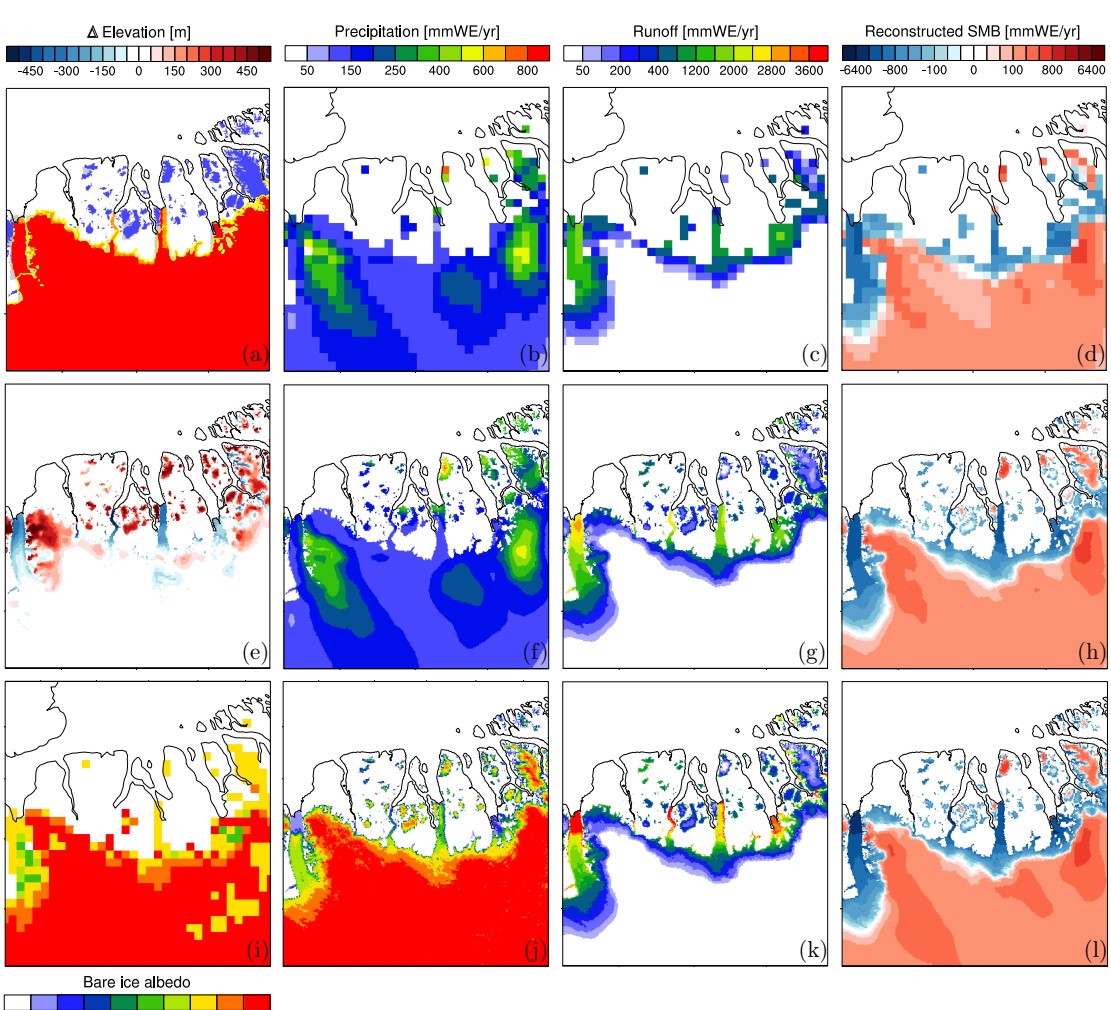

**Fig. 12.** Same as Fig. 9 but for north Greenland (blue box 4 in Fig. 1). The yellow line in a) shows the grounded ice mask at 1-km.