# Peer review of "A daily, 1-km resolution dataset of downscaled Greenland ice sheet surface mass balance (1958-2015)"

_The Cryosphere, 2016_

## Referee Comment (RC1) · Anonymous Referee #1 · 11 Jul 2016

The main purpose of this paper is to present a new 1-km resolution RACMO2.3 dataset, describing the methods used and the successes and failures of the new data, particularly in comparison to the previous standard 11-km RACMO2.3 output. Certainly a 1 km dataset would be widely used and a great asset to the research community.

Overall, I think the authors need to do a better job of making the readers job easy. A reader is going to be interested in this paper to learn about the 1km product, the advantages and disadvantages it has, and how and when to trust the data. To do this, it's important that the authors do more to translate the technical methods into explanations of actual physical processes and mechanisms. Several examples are mentioned more explicitly below, but a primary example is the sentence at 367: "The extreme eleva-

tional SMB gradient that results over the narrow ablation zone is then poorly captured at 11-km, and hence also poorly represented at 1-km." This sentence raised a big red flag in my mind. Isn't the whole point of this downscaling that it can succeed at better representing things like a narrow ablation zone? With better explanation this might not seem so surprising or at least allay doubts the sentence raises about the broader success of downscaling.

The explanation of some of the downscaling methods is not clear. For example, around line 158, the requirement of 6 adjacent ice-covered pixels is confusing. Please clarify: will a pixel only be considered if at least 6 of its neighbor points are ice-covered? In section 3.2, you've explained by 6 points are chosen, but need to provide an explanation for how you decide which 6 points to use. Also, the wording through here can be confusing – e.g., in line 168, does this refer to the 11km or the 1km data?

Comments by line number:

~231-255. This section needs more explanation and translation. The job of the authors is to make the readers job easy. I also thought that the fscale value seems large. Could you add more comment on this? Speculation on how much this influences results? A reader is trying to assess how "true" the results from RACMO2.3 1km product are and some additional comment is needed here to make this clear.

206-207. Can you please provide a citation or additional information about why this is a reasonable assumption?

209. Why is this correction only applied when there's both surface runoff and melt? Shouldn't this be corrected for even when there's only melt because it may actually push the conditions at that pixel into a runoff regime instead of just refreezing or no runoff? Seems like this would influence the correct modeling of runoff extent.

285. How are the climate conditions at Helheim "peculiar"? I'm not aware of any research that discusses conditions there being particularly different than many other

outlet glaciers. This needs further explanation and/or citation.

297+. This is another area where more explanation would be helpful. Rather than just stating that the bias was removed, can you provide any explanation for the mechanisms possibly responsible for this bias? A justification for this adjustment beyond just making the data fit? This is an important opportunity to build confidence in the model and understanding of what the model does/does not do.

325+. Add % in parantheses with all numbers since you started to do this in this section.

448. The end of the paper does not provide a useful summary conclusion. A useful conclusion would provide a final plain language assessment of the 1km product, the main focus of the paper.

Figure 1. Recommend adding legend to figure for yellow and white points.

Figure 4. This figure is confusing. One simple improvement for a) would be changing the ends of the lines that the arrows are on – the arrow should point from the description to the item it's describing. Also, you should 8 sample points here even though the methods indicate 6 were used – the figure should be as close to the actual case as possible. The panel b was poorly referenced in the text and also confusing.

Figure 7. I found it difficult to compare these plots because they all have different axes. How does the agreement look when all put on the same axes? The location labeling in Figure 1 is poor – it's not entirely clear where these transects are and they should probably be represented by something other than overlapping yellow dots. 79N and Helheim have the same agreement problems near the terminus, but only 1 is mentioned in the text.

Figure 8. How well-paired are these? Since it's the agreement between blue and red that is part of the main point, it's important to have a sense about the how well the points pair with each other. The upper right points seem to suggest that it's quite close

– is this similar for all points?

Figure 9. Panel labels should be added since these cover such a wide range of variables/values.
[Figure]

---

## Referee Comment (RC2) · Anonymous Referee #2 · 18 Jul 2016

Review of "A daily, 1-km resolution dataset of downscaled Greenland ice sheet surface mass balance (1958-2015)" by Noël et al., published in *The Cryosphere Discussions*.

**Review Summary:**

This manuscript presents a newly developed downscaled version of the RACMO regional climate/SMB model. The downscaling procedure applies a statistical elevation correction using the GIMP DEM, a correction leading to increased runoff via lower-than-modeled and higher resolution MODIS albedo, and a bias correction to account for RACMO's apparent systematic underestimation of precipitation in the GrIS accumulation zone. The resulting 1-km downscaling of RACMO2.3 shows a notably improved agreement with in situ observations. In particular, the authors highlight better representation of marginal GrIS regions, where complex topography and steep elevation gradients result in large SMB gradients that are poorly resolved in the native 11-km RACMO. Given that RACMO is widely used in the glaciological community, the improved and higher resolution version presented in this paper will certainly be of wide interest.

Below are comments that pertain mostly to the presentation of the material, as opposed to the downscaling methods employed. Overall, I believe the authors need to do a better job at describing the methods and presenting the results in their figures. Finally, I would suggest that the authors consider including more information (perhaps in a new section) on where the new dataset really shows promise for understanding the SMB variability and physical/climatic processes affecting the GrIS. Improved overall agreement with observations is shown (Fig 6), as well as improvements along some transects (Fig 7), and then the authors show some example regions while noting the effects of downscaling. However, the reader is largely left to decipher where the "old" RACMO still works, where the downscaling does a more realistic job, and areas where the SMB is still uncertain (and why). Given the wide use of RACMO, understanding a bit more where these uncertainties lie would be very helpful for the community, while also making for a much stronger manuscript.

**Specific comments:**

Abstract:

You reference the elevation correction almost exclusively here (except for the last sentence), but the albedo correction in particular, and accumulation zone precipitation bias correction to a smaller degree, are also important to improving agreement with the observations. I would add reference to these other two important steps in the abstract.

Figure 4a and text near line 160:

The method for calculating the regression slope ($b_{11km}$) using the adjacent grid cells

is clear: i.e., using a maximum of the 8 adjacent grid cells with different elevations and SMB components to generate a linear regression. This figure (and the text), however is somewhat confusing at first because it looks like a separate regression (the blue line, seemingly labeled a11km) is being generated using only the single blue "current grid-cell" point. After studying this figure, my interpretation is that the blue line is simply applying the (red) regression slope (b11km, red line) and the blue point's x and y values to estimate the intercept, a11km. At first glance, this is not apparent.

I would suggest revising this figure to make it more clear. Perhaps you could make the blue line a different color (e.g., green), make it dotted, and only extending from the blue "current grid-cell" point to the y-axis, and have an arrow pointing toward this intercept point, labeled a11km. This would make it look like less of a separate regression, and more clear that you're just calculating the intercept using the local regression slope (b) and only the central grid cell elevation (x) and SMB component value (y). Perhaps you could also indicate this by showing the formula a11km = y – b11km * x on the figure.

I would also suggest including a legend to label the red points "adjacent grid cell**s**" and blue point "current/central grid point".

Also, you should probably only show 6 adjacent grid cells, as this is what you end up using.

A perhaps bigger question I have about the method is why not just use the intercept value obtained from the first linear regression of the current grid cell and its adjacent cells?

Lines 165-168:

The method step here is unclear. Are you enlarging the native 11-km grid to match the larger spatial extent of the 1-km grid? Can you indicate how often this was used? Is this just at the lateral margins of the ice sheet?

Lines 207-210:

Why not apply this correction to all grid cells? Certainly for some grid cells, this would provide enough additional energy to generate melt (i.e., in non-melt cells), and for other cells, sufficient additional melt to generate runoff in cells where melt doesn't already exceed refreezing. It seems to me that this extra SW absorption may be important not just for cells that already have runoff, but the whole GrIS SEB and snowpack temperatures as well through both increased SW absorption and latent heat release upon refreezing in areas that melt.

Line 248:

Could it also be possible that this is due to only selectively applying the correction to cells already experiencing runoff?

Lines 273-274 / Figure 6a,b:

A more minor point: I calculate a 16.25% decrease in the RMSE, not 18%. On line 279, I calculate a 81.25% decrease in bias, not 88%.

Line 291/Figure 8:

Text here refers to SMB v0.2, but figure states it shows SMB v1.0. Which is it? And you state you applied the bias correction to places where SMBv1.0 > 0 mmWE/yr, but to get SMBv1.0, was it not necessary to first calculate PRv1.0? This seems circular to me.

Figures 9-12:

Can you add the names of these areas from table 1 in the respective figure captions? This would facilitate interpretation and cross reference between figure 1, table 1, sections 5.1-5.5, and these figures. Also, I found myself constantly needing to refer to the caption to interpret the panels. This became more problematic with figures 10-12 as I had to refer back to figure 9's caption. Please add some basic identifiers/titles to each panel.

Panels k and l also incorporate the SMB bias correction, right? This is not clear from the caption, but it is suggested in the text (lines 330-331). Please also specify in the caption.

Lines 334-335:

A systematic overestimation of bare ice albedo is difficult to see here given the different grid sizes. Can you show a plot of the albedo bias as you've done for elevation in panel e of figure 9?

Lines 351-352:

Should these figure references to runoff be Fig 10 g and k (not h and l)?

Line 360:

Similar to above, should these references to increased melt refer to the runoff plots (g and k) rather than SMB (where differences are less perceptible)? Same for line 374.

Lines 414-416:

This affects the entire GrIS, right? Perhaps change "high latitudes" to specify GrIS margins.

Lines 417-418:

I assume the underestimation of bare ice albedo prior to 2000 is because the MODIS time period used of 2000-2015 was one of very high melt, right? If so, you should explicitly state this. This leads me to a second point on the use of the MCD43A3 dataset. It is known that the MODIS Terra sensor has degraded, giving too strong of an albedo decrease for Greenland. The MCD43A3 data are affected since they incorporate both Terra and Aqua observations (e.g., Polashenski et al., 2015, GRL, and others). I would suggest at least acknowledging this as a limitation in this section.

**Technical corrections:**

[Figure]

9: "confined glaciated areas" is a bit unclear, could you reword this somehow?

163: change erratic to erroneous

179: remove comma

181: Specifying that you are estimating b11km would be helpful here.

187: remove "are"

191: specify "native resolution" or 11-km when referring to RACMO2.3 here.

281: specify Figure 11a for yellow dot.

288-290: Please add reference and better explain how this seasonality is different than that of other sites, e.g., Nordbogletscher, which is in a similar region that presumably experiences similar SMB seasonality.

Figure 7 caption: make reference to Figure 1 for locations.

356: remove "the" in "larger the glaciated"

385: Fix reference to Figure 12 i and j.

386: Fix reference to Figure 12 h and l.

---

## Editor Comment (EC1) · L. Koenig (Editor) · 9 Aug 2016

Dear Authors, You have received two detailed reviews with specific items to be addressed in your paper. Please pay particular attention to the point brought up by both reviewers to more clearly state the benefits and potential errors with the new downscaled model run over previous model runs as this will certainly add to the paper and the usability of the data. I look forward to receiving your comments to the reviews at your earliest convenience.

---

## Author Comment (AC1) · 23 Aug 2016

Dear Referee#1, please find attached our response and a revised manuscript.

Sincerely,

Brice Noël

Please also note the supplement to this comment:
http://www.the-cryosphere-discuss.net/tc-2016-145/tc-2016-145-AC1-supplement.pdf

---

## Author Comment (AC2) · 23 Aug 2016

**Authors response:**

Dear reviewers and editor,

To start with, we would like to thank you for your useful and constructive comments. Below, our responses to the individual reviewers' comments are displayed in blue and modifications in the manuscript in orange to facilitate readability.

**Main modifications:**

To address the Reviewers' suggestions, we changed the title of Section 6 to *"Added value, limitations and uncertainties"* and included a new paragraph describing Figure 13 (see additional figure below). In this paragraph we discuss the added value of SMB v1.0 and how it improves on RACMO2.3 when compared to in-situ measurements. SMB v1.0 is an overall improvement on RACMO2.3 but most notably so in the lower ablation zone where the largest elevation and bare ice albedo corrections are applied. *"The downscaled SMB v1.0 is the first dataset to provide daily SMB estimates for all outlet glaciers of the GrIS at a 1-km resolution and for 58 years (1958-2015). Relative to the original RACMO2.3 output, this dataset improves local SMB values (Fig. 7) and produces more realistic SMB patterns over rugged glaciated areas along the GrIS margins (Figs. 9-12). Figs. 6 and 8 show that SMB v1.0 is an overall improvement on the original RACMO2.3. To further investigate this, Fig. 13 shows the annual mean SMB RMSE (model vs. observations) of the 11-km SMB field in RACMO2.3 (red), the downscaled product v0.2 (green) and v1.0 (blue) as a function of observed SMB, binned in 0.5 m w.e. intervals. In the ablation zone (SMB < 0), the SMB RMSE is reduced by 29-65% in v1.0 relative to the 11-km product, owing to the elevation correction in v0.2 (9-23%) and the additional albedo correction (20-42%). In the accumulation zone, the elevation dependence (9%) and the precipitation adjustment (19%) also contribute to reduce the SMB RMSE by 28% in v1.0. The largest RMSE reduction occurs in the lower GrIS ablation zone, where improvements in topography and bare ice albedo in v1.0 are greatest."*

[Figure]

*Fig. 13 Annual mean model SMB RMSE (model vs. observations) of the 11-km SMB field in RACMO2.3 (red dots), the downscaled SMB dataset v0.2 (green dots) and v1.0 (blue dots) as a function of observed SMB (395 observations). Modelled SMB is grouped in 0.5mWE/yr bins except for the first bin, which ranges from -6.00 to -3.75mWE/yr. Numbers indicate the amount of observations used in each bin.*

We also improved the general figure display and modified the manuscript as suggested by the Reviewers. Fig. 6a-c were revised to correct for a small remaining mistake in the associated script. Related numbers in Section 4 were also corrected accordingly.

**Referee#1: Anonymous**

The main purpose of this paper is to present a new 1-km resolution RACMO2.3 dataset, describing the methods used and the successes and failures of the new data, particularly in comparison to the previous standard 11-km RACMO2.3 output. Certainly a 1-km dataset would be widely used and a great asset to the research community.

**General comments:**

Overall, I think the authors need to do a better job of making the readers job easy. A reader is going to be interested in this paper to learn about the 1-km product, the advantages and disadvantages it has, and how and when to trust the data. To do this, it's important that the authors do more to translate the technical methods into explanations of actual physical processes and mechanisms. Several examples are mentioned more explicitly below, but a primary example is the sentence at **L367**: "The extreme elevational SMB gradient that results over the narrow ablation zone is then poorly captured at 11-km, and hence also poorly represented at 1-km." This sentence raised a big red flag in my mind. Isn't the whole point of this downscaling that it can succeed at better representing things like a narrow ablation zone? With better explanation this might not seem so surprising or at least allay doubts the sentence raises about the broader success of downscaling.

We removed the sentence at **L367**, as it was unclear. Even at this location, most of the remaining bias that persists after the elevation correction in v0.2 (Fig. 6b) is removed in v1.0 when correcting for ice albedo (Fig. 6c).

The explanation of some of the downscaling methods is not clear. For example, around **L158**, the requirement of 6 adjacent ice-covered pixels is confusing. Please clarify: will a pixel only be considered if at least 6 of its neighbor points are ice-covered? In section 3.2, you've explained by 6 points are chosen, but need to provide an explanation for how you decide which 6 points to use.

Again, the original text was unclear. We use the maximum amount of points available to calculate a regression but at least 6 cells are required, i.e. the actual grid point and at least 5 adjacent pixels. These grid cells must 1) be adjacent to the current pixel, 2) be ice-covered, 3) have a non-zero value when downscaling melt and runoff. We apply condition 3) to preclude null or extreme runoff/melt regression slopes. If the current grid cell does not obey all these conditions, it is discarded until the extrapolation step. To make this clear, we reformulated as follow: "[…] ice-covered RACMO2.3 grid-point *using the maximum amount of points available, i.e. we use a total of six to nine ice-covered grid cells, being the current one and minimum five to maximum eight adjacent pixels.*"

Also, the wording through here can be confusing – e.g., in **L168**, does this refer to the 11km or the 1km data?

Here we refer to the 11-km grid, as was stated in the manuscript. At this stage, the downscaling algorithm extrapolates outwards - on the 11-km grid - the values obtained during the previous step of the regression slopes ($b_{11km}$) and intercepts ($a_{11km}$) to fully cover the 11-km domain and hence the entire ice mask at 1-km.

**Specific comments:**

**L231-255**.
**1)** This section needs more explanation and translation. The job of the authors is to make the readers job easy.

To stress the aim of the ice albedo correction, we included the following sentence: *"The bare ice albedo bias correction aims at minimizing the misfit between downscaled SMB v0.2 and in-situ measurements (Fig. 6b) by estimating the missing runoff in the low ablation zone in v0.2."*

**2)** I also thought that the fscale value seems large. Could you add more comment on this? Speculation on how much this influences results?

To clarify the meaning and impact of $f_{scale}$, we added the following lines: *"This means that $RU_{add}$, i.e. accounting for elevation and bare ice albedo corrections, has yet to be increased by ~18% to optimise the agreement between downscaled and in-situ SMB (Fig. 6c)."*

In the manuscript, we explicitly stress that underestimated sensible heat fluxes (Fausto et al. [2016]) and underestimated cloud formation at the GrIS margins (Van Tricht et al. [2015]) are most likely responsible for the ~18% underestimation in ablation. However, the statistical downscaling approach is not designed to correct for these physical processes. We inserted the following sentence: *"However, as the statistical downscaling approach is not designed to correct for these physical processes, we adopted the empirical approach presented above."*

**3)** A reader is trying to assess how "true" the results from RACMO2.3 1km product are and some additional comment is needed here to make this clear.

See Main Modifications above and the additional Figure 13.

**L206-207**. Can you please provide a citation or additional information about why this is a reasonable assumption?

There is no physical basis for this assumption, which can be further refined in future versions of the downscaling procedure. The partitioning of diffuse and direct radiation is highly sensitive to weather conditions, i.e. clear or overcast sky, and can be highly variable within a single day (cloud formation or advection). Because RACMO2.3 output does not provide this partitioning on a daily basis, and to keep the approach straightforward, we decided to equally partition total radiation into direct and diffuse radiation. To clarify, we added the following sentence: *"This assumption is purely pragmatic; on the basis of data availability, it could be further refined in future versions of the downscaling procedure."*

**L209**.
**1)** Why is this correction only applied when there's both surface runoff and melt?

The reasoning here is that the ablation underestimation in Fig. 6b is mostly driven by overestimated ice albedo in the 11-km SMB product. Therefore, the correction is exclusively applied in the ablation zone (SMB < 0) characterized by non-zero melt and runoff in v0.2. We inserted the following sentence: *"Figs. 6b and 8 show that ablation underestimation in v0.2 is restricted to the low ablation zone (SMB < -4 mWE), where bare ice is exposed for long episodes in summer. Therefore, the following corrections are only applied to the ablation zone on days of melting bare ice when both surface runoff and melt are non-zero in the downscaled product v0.2:"*

**2)** Shouldn't this be corrected for even when there's only melt because it may actually push the conditions at that pixel into a runoff regime instead of just refreezing or no runoff? Seems like this would influence the correct modeling of runoff extent.

The melt/runoff area is already corrected by applying the elevation dependence in v0.2 and further reconstructing melt and runoff using the estimated gradients and topography at 1-km. As a result, SMB v0.2 improves much the representation of the ablation zone although significant SMB biases remain and are corrected in v1.0.

**L285**. How are the climate conditions at Helheim "peculiar"? I'm not aware of any research that discusses conditions there being particularly different than many other outlet glaciers. This needs further explanation and/or citation.

The Helheim transect shows very small measured SMB gradients compared to other locations (see 'b' value in the plots), resulting in a quasi-constant ablation rate for all elevations (~ -1 mWE). The reason for this low SMB gradient is not clear at present although we noticed that each individual observation covers only 1 or 2 summer months. We reformulated as follow: *"The downscaled product fails at reproducing the quasi-constant ablation rate (~ -1 mWE) characterizing the Helheim transect. The reason for this low SMB gradient is not clear at present; it may be due to uncertainties in individual observation covering relatively short periods, i.e. 1 or 2 months, which are only limited to the melt season (July-August). Another possible explanation is that Helheim glacier experiences pronounced winter accumulation at low elevations, potentially caused by drifting snow transport, limiting summer ablation."*

**L297+**.
**1)** This is another area where more explanation would be helpful. Rather than just stating that the bias was removed, can you provide any explanation for the mechanisms possibly responsible for this bias?

In RACMO2.3, clouds formation occurs at too low elevations resulting in overestimated precipitation at the margins, e.g. southeast Greenland, while the ice sheet interior experiences too dry conditions. This bias is partly solved in RACMO2.3 by allowing ice cloud super-saturation, which delays precipitation formation to higher elevations (Noël et al., 2015). To overcome this bias, additional improvements have to be implemented in the cloud scheme. Eventually, repeating the downscaling procedure using forthcoming RACMO2.4 output would make the precipitation adjustment unnecessary. We inserted the following sentence: *"Despite significant improvements in the cloud scheme of RACMO2.3* (Noël et al., 2015)*, clouds become saturated and start to produce precipitation at too low elevations, resulting in overestimated precipitation at the margins, e.g. southeast Greenland, while the ice sheet interior experiences too dry conditions. This precipitation bias is currently being investigated, and we aim to resolve it in the upcoming version RACMO2.4."*

**2)** A justification for this adjustment beyond just making the data fit? This is an important opportunity to build confidence in the model and understanding of what the model does/does not do.

The justification is that RACMO3 systematically underestimated accumulation in the interior, because of the process described above. We added: *"To overcome the systematic negative SMB bias of RACMO2.3 in the GrIS accumulation zone (-37.5 mmWE/yr, Fig. 8), the daily total precipitation v0.2 is adjusted to correct for underestimation in the ice sheet accumulation zone (SMB > 0 mmWE/yr):"*

**L325+**. Add % in parentheses with all numbers since you started to do this in this section.

Thank you for pointing that out.

**L448**. The end of the paper does not provide a useful summary conclusion. A useful conclusion would provide a final plain language assessment of the 1km product, the main focus of the paper.

We added the following paragraph at the end of the discussion section: "*We anticipate that the new, 1-km Greenland SMB product is especially useful for studies that address the mass balance of Greenland outlet glaciers that are too steep and/or narrow to be properly resolved at the typical horizontal resolution of regional climate models (~ 5-15 km). Future downscaled products can have even higher resolution (100m) and will be based on further improved RCM output fields of precipitation and melt.*"

**Figure 1**. Recommend adding legend to figure for yellow and white points.
We included the suggested legend accordingly.

**Figure 4**. This figure is confusing. One simple improvement for a) would be changing the ends of the lines that the arrows are on – the arrow should point from the description to the item it's describing. Also, you should 8 sample points here even though the methods indicate 6 were used – the figure should be as close to the actual case as possible. The panel b was poorly referenced in the text and also confusing.
We decided to follow Referee#2 suggestions to improve Fig. 6a (see Referee#2 Figure 4a). We decided to keep Fig. 4b as is and now refer to it earlier in the text.

**Figure 7**.
**1)** I found it difficult to compare these plots because they all have different axes. How does the agreement look when all put on the same axes?
We decided not to modify the axes ranges. The axis range was chosen for each transect to optimize the readability and clarity of the figure. Using identical axes would prevent clear distinction between downscaled and in-situ SMB for many transects.

**2)** The location labeling in Figure 1 is poor – it's not entirely clear where these transects are and they should probably be represented by something other than overlapping yellow dots.
Owing to the large number of measurements, the multiple case study regions and the transects' names, it would be somewhat difficult to improve the overall display of this figure. We deemed it important to show as many stations as possible even though they locally overlap. Letters are now closer to the associated transect to improve clarity.

**3)** 79N and Helheim have the same agreement problems near the terminus, but only 1 is mentioned in the text.
Thank you for pointing this out. We hypothesize that at 79N and Storstrømmen, SMB measurements lower than 200 m are located on floating glacier tongues with melt ponds, resulting in very low satellite albedo, while stake measurements are performed between ponds on brighter surfaces. As a result, the bare ice albedo correction could be overestimated. We now discuss this in more detail in the revised text: *"In addition, Nioghalvjerds-fjorden and Storstrømmen transects (Figs. 7a-b) also show significant remaining biases between in-situ and downscaled SMB at elevations lower than 200 m. We hypothesize that these SMB measurements are located on floating glacier tongues with melt ponds, resulting in very low satellite albedo, while stake measurements are performed between ponds on brighter surfaces. As a result, the bare ice albedo correction could be overestimated."*

**Figure 8**. How well-paired are these? Since it's the agreement between blue and red that is part of the main point, it's important to have a sense about the how well the points pair

with each other. The upper right points seem to suggest that it's quite close, is this similar for all points?

The aim of this figure is to show that the downscaled product v0.2 (blue dots) generally fits the observations better (dashed black line) than the original RACMO2.3 output (red dots). This is also supported by slightly reduced bias and RMSE in the downscaled product v0.2. Ultimately, the bias of course becomes zero in v1.0.

**Figure 9**. Panel labels should be added since these cover such a wide range of variables/ values.

We modified the figure display accordingly (see Referee#2 Figures 9-12).

**Referee#2: Anonymous**

This manuscript presents a newly developed downscaled version of the RACMO regional climate/SMB model. The downscaling procedure applies a statistical elevation correction using the GIMP DEM, a correction leading to increased runoff via lower than-modeled and higher resolution MODIS albedo, and a bias correction to account for RACMO's apparent systematic underestimation of precipitation in the GrIS accumulation zone. The resulting 1-km downscaling of RACMO2.3 shows a notably improved agreement with in situ observations. In particular, the authors highlight better representation of marginal GrIS regions, where complex topography and steep elevation gradients result in large SMB gradients that are poorly resolved in the native 11-km RACMO. Given that RACMO is widely used in the glaciological community, the improved and higher resolution version presented in this paper will certainly be of wide interest.

**General comments:**

Below are comments that pertain mostly to the presentation of the material, as opposed to the downscaling methods employed. Overall, I believe the authors need to do a better job at describing the methods and presenting the results in their figures. Finally, I would suggest that the authors consider including more information (perhaps in a new section) on where the new dataset really shows promise for understanding the SMB variability and physical/climatic processes affecting the GrIS. Improved overall agreement with observations is shown (Fig 6), as well as improvements along some transects (Fig 7), and then the authors show some example regions while noting the effects of downscaling. However, the reader is largely left to decipher where the "old" RACMO still works, where the downscaling does a more realistic job, and areas where the SMB is still uncertain (and why). Given the wide use of RACMO, understanding a bit more where these uncertainties lie would be very helpful for the community, while also making for a much stronger manuscript.

See Main Modifications above and the additional Figure 13.

**Specific comments:**

**Abstract**. You reference the elevation correction almost exclusively here (except for the last sentence), but the albedo correction in particular, and accumulation zone precipitation bias correction to a smaller degree, are also important to improving agreement with the observations. I would add reference to these other two important steps in the abstract.

We reformulated as: *"Applying corrections for elevation, bare ice albedo and accumulation bias, the high-resolution product is statistically […] RACMO2.3 at 11-km."*

**Figure 4a and L160+**. The method for calculating the regression slope (b11km) using the

adjacent grid cells is clear: i.e., using a maximum of the 8 adjacent grid cells with different elevations and SMB components to generate a linear regression. This figure (and the text), however is somewhat confusing at first because it looks like a separate regression (the blue line, seemingly labeled a11km) is being generated using only the single blue "current grid-cell" point. After studying this figure, my interpretation is that the blue line is simply applying the (red) regression slope (b11km, red line) and the blue point's x and y values to estimate the intercept, a11km. At first glance, this is not apparent. I would suggest revising this figure to make it more clear. Perhaps you could make the blue line a different color (e.g., green), make it dotted, and only extending from the blue "current grid-cell" point to the y-axis, and have an arrow pointing toward this intercept point, labeled a11km. This would make it look like less of a separate regression, and more clear that you're just calculating the intercept using the local regression slope (b) and only the central grid cell elevation (x) and SMB component value (y). Perhaps you could also indicate this by showing the formula a11km = y – b11km * x on the figure. I would also suggest including a legend to label the red points "adjacent grid cells" and blue point "current/central grid point". Also, you should probably only show 6 adjacent grid cells, as this is what you end up using.

Thank you for these useful suggestions, we modified Fig. 4a accordingly (see updated figure below).

[Figure]

A perhaps bigger question I have about the method is why not just use the intercept value obtained from the first linear regression of the current grid cell and its adjacent cells?

Applying the regression directly would result in regionally smoothened a11km and a1km fields, negatively impacting the added value of the 1-km SMB components. To highlight this we inserted the following sentence: *"The regression is applied to the current grid cell to prevent local estimates of a11km to significantly differ from the original RACMO2.3 value."*

**L165-168**.

1) The method step here is unclear. Are you enlarging the native 11-km grid to match the larger spatial extent of the 1-km grid?

Indeed. To make this clear, we reformulated as: *"Next, valid estimates […] on the 11-km grid to fully cover the more extensive 1-km ice mask."*

2) Can you indicate how often this was used?

The extrapolation is also daily specific; we now mention it clearly in the manuscript. *"To that end daily regression parameters are extrapolated outwards […] ."*

3) Is this just at the lateral margins of the ice sheet?

The algorithm extrapolates a11km and b11km at the ice sheet margins as the relatively coarse

11-km RACMO2.3 ice mask is not able to resolve small Greenland peripheral ice caps and marginal GrIS glaciers for which estimates of $a_{1km}$ and $b_{1km}$ are crucial to reconstruct SMB at 1-km.

**L207-210**. Why not apply this correction to all grid cells? Certainly for some grid cells, this would provide enough additional energy to generate melt (i.e., in non-melt cells), and for other cells, sufficient additional melt to generate runoff in cells where melt doesn't already exceed refreezing. It seems to me that this extra SW absorption may be important not just for cells that already have runoff, but the whole GrIS SEB and snowpack temperatures as well through both increased SW absorption and latent heat release upon refreezing in areas that melt.
See response to Referee#1 at **L209 1-3**.

**L248**. Could it also be possible that this is due to only selectively applying the correction to cells already experiencing runoff?
The $f_{scale}$ value is mostly driven by the missing ablation in low-lying areas where sensible heat flux may be underestimated in RACMO2.3 (Noël et al. (2015) and Fausto et al. (2016)). The ablation zone is well represented in the downscaled product owing to 1) the elevation correction of runoff (v0.2); 2) the bare ice albedo correction in ablation areas showing a surface albedo < 0.55, i.e. bare ice exposure at the surface. See also reply to Referee#1 L209 (2) and L231-255 (2).

**L273-274 and Figures 6a,b**. A more minor point: I calculate a 16.25% decrease in the RMSE, not 18%. On **L279**, I calculate a 81.25% decrease in bias, not 88%.
Thank you for pointing this out. As explained in Main Modifications, Fig. 6a-c still included small mistakes and had to be revised. We also corrected the associated numbers accordingly.

**L291 and Figure 8**. Text here refers to SMB v0.2, but figure states it shows SMB v1.0. Which is it?
It should be v0.2 instead of v1.0, thank you.

And you state you applied the bias correction to places where SMBv1.0 > 0 mmWE/yr, but to get SMBv1.0, was it not necessary to first calculate PRv1.0? This seems circular to me.
This was unclear in the original text. We reformulated as: *"To overcome the systematic negative SMB bias of RACMO2.3 in the GrIS accumulation zone (-37.5 mmWE/yr, Fig. 8), the daily total precipitation v0.2 is adjusted to correct for underestimation in the ice sheet accumulation zone (SMB > 0 mmWE/yr):"*

We also added the following sentence: *"The final SMBv1.0 product is reconstructed as: SMBv1.0 = PRv1.0 - RUv1.0 -SU - ER (12)"*

Figures 9-12. Can you add the names of these areas from table 1 in the respective figure captions? This would facilitate interpretation and cross reference between figure 1, table 1, sections 5.1-5.5, and these figures. Also, I found myself constantly needing to refer to the caption to interpret the panels. This became more problematic with figures 10-12 as I had to refer back to figure 9's caption. Please add some basic identifiers/titles to each panel.
We updated Figs. 9-12 and captions accordingly (see updated Fig. 9 at **L334-335** below).

Panels k and l also incorporate the SMB bias correction, right? This is not clear from the caption, but it is suggested in the text (lines 330-331). Please also specify in the caption.

We updated the figures and modified the caption accordingly. "**Centre east:** a) Ice sheet mask in RACMO2.3 at 11-km (red) and in the down-sampled GIMP DEM at 1-km (orange) (blue box 1 in Fig. 1), and the mask of disconnected glaciers and ice caps at 1-km (blue); average (1958-2015) annual mean b) total precipitation, c) runoff and d) SMB (mmWE/yr) modelled by RACMO2.3 at 11km; e) elevation bias (m) between 1-km and 11-km resolutions. Figures f), g), h) represent annual mean total precipitation, runoff and SMB downscaled to 1-km using elevation dependence only (v0.2). Figure i) shows the bare ice albedo bias between MODIS measurements at 1-km (2000-2015) and RACMO2.3 at 11-km (2001- 2010). Figures j), k) and l) are similar to f), g) and h) but incorporate the bare ice albedo and precipitation corrections (v1.0)."

**L334-335**. A systematic overestimation of bare ice albedo is difficult to see here given the different grid sizes. Can you show a plot of the albedo bias as you've done for elevation in panel e of figure 9?

As suggested, we replaced Fig. 9-12 (i) by ΔBIA (bare ice albedo) and Fig.9-12 j by PRv1.0 (see example Figure below).

[Figure]

**L351-352**. Should these figure references to runoff be Fig 10 g and k (not h and l)?
Yes it should, thank you.

**L360**. Similar to above, should these references to increased melt refer to the runoff plots (g and k) rather than SMB (where differences are less perceptible)? Same for **L374**.
Thank you for pointing this out, we modified as suggested.

**L414-416**. This affects the entire GrIS, right? Perhaps change "high latitudes" to specify GrIS margins.

This issue mostly occurs at high latitudes where the satellite signal shows a slight tilt, potentially mixing radiances from neighboring tundra and ice scenes. It also affects the glacier floating tongues at the GrIS margins so we inserted the following sentence: "*Note that floating glacier tongues also show too low surface albedo, e.g. Petermann glacier (yellow dot in Fig. 12a), resulting from mixed signals from adjacent dark melt pond and brighter dry ice. The resulting albedo underestimation over low-lying floating tongues below 200 m leads to overestimated ablation (∼ 0.2 mWE/yr; Figs. 7a and b); […]*".

**L417-418**.
**1)** I assume the underestimation of bare ice albedo prior to 2000 is because the MODIS time period used of 2000-2015 was one of very high melt, right? If so, you should explicitly state this.
We reformulated as: "*[…] underestimate the bare ice albedo prior to 2000 as the period 2000-2015 encompasses multiple record high melt years.*".

**2)** This leads me to a second point on the use of the MCD43A3 dataset. It is known that the MODIS Terra sensor has degraded, giving too strong of an albedo decrease for Greenland. The MCD43A3 data are affected since they incorporate both Terra and Aqua observations (e.g., Polashenski et al., 2015, GRL, and others). I would suggest at least acknowledging this as a limitation in this section.
Thank you for this suggestion, we included this as an additional limitation. "*[…]; d) the degradation of MODIS Terra sensors (Polashenski et al., 2015).*"

**Technical corrections:**

**L9**. "confined glaciated areas" is a bit unclear, could you reword this somehow?
We reworded as "isolated".
**L163**. change erratic to erroneous. OK.
**L179**. remove comma. OK.
**L181**. Specifying that you are estimating b11km would be helpful here. OK.
**L187**. remove "are". OK.
**L191**. specify "native resolution" or 11-km when referring to RACMO2.3 here. OK.
**L281**. specify Figure 11a for yellow dot. OK.
**L288-290**. Please add reference and better explain how this seasonality is different than that of other sites, e.g., Nordbogletscher, which is in a similar region that presumably experiences similar SMB seasonality. See response to Reviewer#1 **L285**.
**Figure 7 caption**. make reference to Figure 1 for locations. OK.
**L356**. remove "the" in "larger the glaciated". OK.
**L385**. Fix reference to Figure 12 i and j. OK.
**L386**. Fix reference to Figure 12 h and l. OK.

[revised manuscript text omitted]